# The effectiveness of organisational-level workplace mental health interventions on mental health and wellbeing in construction workers: A systematic review and recommended research agenda

Birgit A. Greiner[1]*, Caleb Leduc[1,2], Cliodhna O'Brien[2], Johanna Cresswell-Smith[3], Reiner Rugulies[4,5], Kristian Wahlbeck[3], Kahar Abdulla[6], Benedikt L. Amann[7,8,9,10], Arlinda Cerga Pashoja[11,12], Evelien Coppens[13], Paul Corcoran[2], Margaret Maxwell[14], Victoria Ross[15], Lars de Winter[16], Ella Arensman[1,2,17,18], Birgit Aust[4]

1 School of Public Health, University College Cork, Cork, Ireland, 2 National Suicide Research Foundation, University College Cork, Cork, Ireland, 3 Finnish Institute for Health and Welfare, Helsinki, Finland, 4 National Research Centre for the Working Environment, Copenhagen, Denmark, 5 Department of Public Health, University of Copenhagen, Denmark, 6 European Alliance Against Depression e.V., Frankfurt, Germany, 7 Centre Fòrum Research Unit, Institut de Neuropsiquiatria i Addiccions (INAD), Parc de Salut Mar, Barcelona, Spain, 8 Hospital del Mar Medical Research Institute (IMIM), Barcelona, Instituto Carlos III, CIBERSAM, Madrid, Spain, 9 Department of Psychiatry and Psychotherapy, Klinikum der Universität, München, Munich, Germany, 10 Universitat Pompeu Fabra, Barcelona, Spain, 11 London School of Hygiene and Tropical Medicine, London, England, 12 Faculty of Sport, Allied Health and Performance Science, St Marys University, London, England, 13 LUCAS, Centre for Care Research and Consultancy, KU Leuven, Leuven, Belgium, 14 The Nursing, Midwifery and Allied Health Professions Research Unit, University of Stirling, Stirling, Scotland, 15 Australian Institute for Suicide Research and Prevention, Griffith University, Griffith, Queensland, Australia, 16 Phrenos Center of Expertise for Severe Mental Illnesses, Utrecht, The Netherlands, 17 Australian Institute for Research, Griffith University, Mount Gravatt, Australia, 18 International Association for Suicide Prevention, Washington, DC, United States of America

* b.greiner@ucc.ie

## Abstract

### Objectives

This systematic review assesses the scientific evidence regarding the effectiveness of organisational-level workplace mental health interventions on stress, burnout, non-clinical depressive and anxiety symptoms, and wellbeing in construction workers.

### Methods

Eligibility criteria were randomized controlled trials (RCTs), cluster randomized controlled trials (cRCTs), controlled or uncontrolled before- and after studies published in peer-reviewed journals between 2010 and May 2022 in five databases (Academic Search Complete, PsycInfo, PubMed, Scopus and Web of Science). Outcomes were stress, burnout and non-clinical depression and anxiety symptoms, and wellbeing (primary) and workplace changes and sickness absenteeism (secondary). Quality appraisal was conducted using the QATQS scale, a narrative synthesis was applied. The protocol was published in PROSPERO

**Data Availability Statement:** All relevant data are within the paper and its Supporting Information files.

**Funding:** Funding This study is funded by the European Union's Horizon 2020 research and innovation programme under grant agreement No. 848137. The material presented and views expressed here are the responsibility of the author (s) only. The EU Commission takes no responsibility for any use made of the information set out. The funder did not play any role in the study design, data collection and analysis, decision to publish, or preparation of the manuscript.

**Competing interests:** The authors have declared that no competing interests exist.

CRD42020183640 https://www.crd.york.ac.uk/prospero/display_record.php?ID=CRD42020183640.

## Main results

We identified five articles (four studies) with a total sample size of 260, one cRCT, one controlled before- and after study, and two uncontrolled before- and after studies. The methodological quality of one study was rated as moderate, while for three studies it was weak. One study showed significant effects of a work redesign programme in short-term physiological stress parameters, one study showed a significant employee perceived improvement of information flow after supervisor training and one study showed a substantial non-significant decline in sick leave. There was no significant effect on general mental health (SF12) nor on emotional exhaustion. The focus of all studies was on physical health, while detailed mental health and wellbeing measures were not applied.

## Main conclusions

The evidence for the effectiveness of organisational-level workplace mental health interventions in construction workers is limited with opportunities for methodological and conceptual improvement. Recommendations include the use of a wider range of mental health and wellbeing outcomes, interventions tailored to the specific workplace and culture in construction and the application of the principles of complex interventions in design and evaluation.

## Introduction

Mental health in the workplace is an important area of attention worldwide, owing to its impact on sickness absence, early retirement, and productivity [1,2]. The annual cost of mental ill health to economies has broadly been estimated to be $2.5 trillion globally in 2010; a figure expected to rise to $6 trillion by 2030 [3,4]. Construction is one of the largest employment sectors in the European Union (EU) encompassing 18 million workers, and making a contribution of 9% to the EU's Gross Domestic Product (GDP) [5]. The mental health of construction workers has recently moved to the foreground in occupational health and safety research [6]. Traditionally, research and practice in this sector focussed on safety and physical health owing to its highly hazardous working environment. However, mental health is now increasingly recognised as an area in need of attention [7], and in particular due to the exacerbated challenges presented by the Covid-19 pandemic, that hit the construction sector particularly hard with heightened job insecurity, heavier workloads, social isolation during teleworking and challenges due to home-work practices [8].

Mental health has particular relevance in the safety-sensitive construction industry, as research shows that common mental health disorders are associated with reduced work ability [9] and heightened workplace accident and injury risk [10]. Qualitative research details how mental health symptoms as well as mental health medication can impair performance and safety behaviour [11]. Common mental disorders are prevalent in construction workers as shown in large representative population surveys. For example, the prevalence rate was 17% in skilled male construction workers assessed by diagnostic interview [12], lifetime depression prevalence was 13% in Michigan construction workers[13]. These results were matched with New England construction workers where 16% reported substantial mental distress, follow-up diagnostic interviews revealed that nine out of ten affected workers fulfilled the diagnostic

criteria for a mental disorder [14].There is consistent evidence in the literature on high rates of suicide in the construction sector, particularly in relation to low-skilled workers [6,15,16]. For example, the UK Office of National Statistics data in 2011–2015 showed that low-skilled male construction workers had 3.7 times greater risk of suicide than the national male average, and for skilled-trades workers the suicide risk doubled [6]. Some scholars warned that in light of the COVID-19 pandemic, mental health may deteriorate leading to increased risk of suicide in the construction sector, contributing to what was described as 'a perfect storm' [17].

Furthermore, some studies indicated that mental health problems which do not qualify as mental disorders according to official disease classification, such as stress [7,14,18,19], burnout [20,21] and elevated depressive symptoms [9,19] were also particularly prevalent in the construction sector [22]. Although sometimes confused with clinical mental health disorders, these non-clinical mental health problems constitute distinct concepts. In the context of workplace mental health promotion, they are relevant as they are prevalent in working populations and may constitute risk factors for manifesting mental health and physical disorders [23–25].

## The psychosocial work environment in the construction sector

Characteristics of the psychosocial work environment in the construction sector may be risk factors for poor mental health [26,27]. A challenging feature of construction work is the project- and client-driven nature of the building trade, which results in a high occurrence of short-term projects along temporary contracts, job and financial insecurity, and work away from home for longer periods of time [7,20,26,28]. A systematic review identified high work demands (such as long hours) and lack of job control (e.g., few opportunities for decision-making and opportunities to speak about work issues) as the primary stressors for construction workers [26,27]. According to a meta-analysis, role conflict (incompatible work tasks or expectations), role ambiguity (lack of clarity or uncertainty about job responsibilities), job insecurity, interpersonal conflict (disagreement or frictions) and role overload (excessive workload, long hours, time pressure) were the stress factors which showed the strongest pooled associations with work-related distress and burnout [29]. In an industry that relies heavily on the timely flow of information and efficient communication between different collaborators of a project, poor communication and information sharing were deemed to be fundamental psychosocial risks [30]. Challenges to mental health of workers were reported as being exacerbated for Micro, and Small and Medium Enterprises (SME's), that often have limited financial and personnel resources to engage in psychosocial risk assessment and management or in mental health promotion [31]. Short-term business survival tends to hold priority over occupational health priorities [32,33]. Further challenges specific to SME's relate to financial arrangements as payments are commonly on a project-completion basis and are often late, putting the smaller companies and contractors under pressure to engage their staff in long working hours to ensure cashflow [7].

## Workplace mental health promotion in the construction sector

The importance of mental health in construction workers has been increasingly recognised over the past years, e.g. by a comprehensive report published by the Chartered Institute of Building [7]. The industry established a variety of mental health programmes including training and education for workers and management and provision of services for those in need [34]. MATES in Construction in Australia offers a programme designed to address elevated suicide rates within the construction industry, via trained peer support structures, resources for those in need, and social support structures and community building. Since its inception in 2008, a growing body of research indicates the program is an acceptable and effective

intervention that has significantly contributed to reducing stigmatising beliefs about suicide and mental health, and improving help-seeking and help-offering within the construction sector [35–39]. Similarly, charities like MATES in Mind in the UK (https://www.matesinmind.org/), Construction Working Minds in the US (https://www.constructionworkingminds.org/) and the Lighthouse Construction Industry Charity (https://www.lighthouseclub.org/) provide services, helplines, training and education tailored to the construction industry together with financial and emotional support to workers and their families. Wellbeing and health promotion initiatives in this sector gained momentum when showcased in major construction projects with an explicit ethos to protect and promote both safety as well as health and wellbeing of workers. Prominent examples are the Olympic Park for the London Olympics 2012 [40] and the Heathrow Terminal 5 constructions [41]. Scientifically published intervention studies for construction workers mainly focus on individual-based mental health approaches, such as stress management training [42,43], mindfulness training [44], interventions targeted at individual behavioural change to adapt healthy lifestyles, such as exercising, healthy diet and smoking cessation [45,46] and mental health anti-stigma training [47].

Despite the increasing attention to mental health in this sector, implementation of programmes into the day-to-day operations of organisations appear to be limited. For example, a survey of 1444 Irish and British health and safety professionals registered with the Institution of Occupational Safety and Health (IOSH), showed that those health and safety professionals working in the construction industry were the least likely to be engaged in the development or implementation of psychosocial work prevention programmes (22.4%) compared to other sectors. They were also less likely to engage in broader health promotion actions at work as part of their roles as health and safety professional in the company [48]. Specific structural constraints in this sector pose unique challenges to sustained implementation of workplace mental health promotion programmes [49]. The transient nature of the work, as well as workers being spread out over interchangeable work sites, combined with a high proportion of the workforce being on short-term contracts hampers the reach of sustained health promotion activities [32]. Furthermore, it has also been highlighted that male-dominated workplaces may be subject to traditional masculine values such as self-reliance and stoicism, which limit help-seeking behaviour and create higher levels of mental health-related stigma, requiring specific approaches in terms of mental health promotion [35,50]. Language and literacy challenges as well as cultural differences also need to be taken into consideration within a sector that employs high numbers of migrant workers and a low-educated workforce [51,52].

The growing attention for construction worker health has sparked a critical discussion about the most effective management of health risks. A major point of criticism has been that programmes targeting solely individual coping and behaviour change can draw attention away from addressing the underlying working conditions [53] and putting the responsibility entirely on the worker [54]. Jones at al elaborated "An example of this is the well-intentioned focus on mental health in construction, which encourages workers to identify and seek help for their problems but does not commit organisations or the industry to taking action to reduce the risks inherent in the sector" [54, p.546]. However, recently developed conceptual frameworks to guide practical management and research in construction integrate both individual-level and organisational-level approaches for the protection against workplace mental health risks and the promotion of wellbeing [55–57].

## Towards an integrated mental health promotion approach

The Integrated Workplace Mental Health Intervention approach [57,58] is a widely recognised framework. It integrates individual-based approaches, such as providing support to those with

mental health problems and strengthening coping resources with organisational-level approaches aiming to create sustainable structures, workplace resources, organisational cultures and working conditions. that protect mental health and support positive mental health and wellbeing. It highlights three components for workplace interventions to achieve the utmost mental health benefits: 1) protect mental health by reducing work–related risk factors (psychosocial risks); 2) promote mental health and wellbeing by developing positive aspects of work and worker capacities; and 3) address mental health problems among employees and management regardless of cause. Given a range of published systematic reviews and meta reviews on the effectiveness of individual-level workplace health interventions on mental health and wellbeing outcomes [57,59,60], this review will be focussed on the effectiveness of organisational level mental health and wellbeing interventions.

Organisational-level interventions aim to improve workers' health or wellbeing through the intentional targeting of the organisation and management of work (e.g., task design, psychosocial work conditions, practices of work) or organisational policies and procedures to improve and monitor psychosocial working conditions. They are usually delivered to an entire organisation or units of an organisation [60,61]. Organisational-level interventions support wellbeing by typically addressing the underlying psychosocial factors in the work environment and organisational cultures that lead to job stress, burnout and depressive/anxiety symptoms [62]. A recent systematic review classified four types of organisational- and group-level work interventions for worker wellbeing: 1, changes in scheduling practices (when and where work is completed); 2, job and task changes how work is done; 3, relational and team dynamics interventions (e.g. regular team meetings); and 4, participatory interventions with an emphasis on the change process itself by inviting workers to identify stressors and make changes to work practices or interpersonal relationships [63].

While the effectiveness of organisational-level interventions has been systematically reviewed for a range of mental and physical health indicators and for a range of occupations [63–65], the evidence for the impact of organisational-level mental health interventions in the construction industry and particularly for SMEs in this sector has not been systematically evaluated and published.

## Objectives

This systematic review is part of a series of reviews informing the Horizon 2020 funded European Intervention Project 'Mental Health Promotion in Occupational Settings'(MENTUPP). MENTUPP aims to improve mental health and wellbeing in the workplace by developing, implementing, and evaluating a comprehensive multi-level intervention targeting both clinical and non-clinical mental health difficulties and promoting mental wellbeing in three sectors: healthcare, information and communications technology (ICT) and construction. While a previous review conducted by the MENTUPP Consortium detailed the effects of interventions addressing clinical mental health disorders across all industries [66], this review focuses on non-clinical mental health outcomes (i.e., stress, burnout, and moderately elevated depressive and anxiety symptoms), and to enhance mental wellbeing in the construction sector. Two other reviews about organisational-level interventions focusing on non-clinical mental health and wellbeing in ICT and the health care sectors are currently being prepared. The current review will inform the tailoring of the MENTUPP interventions to the needs of the construction sector.

The specific review questions are: (1) Are organisational-level mental health programmes effective in reducing stress, burnout, non-clinical depressive and anxiety symptoms, and in enhancing mental wellbeing in construction workers? (2) Are organisational-level mental

health programmes effective in reducing stress, burnout, non-clinical depressive and anxiety symptoms and in enhancing mental wellbeing in SME construction workers?

## Materials and methods

The systematic review was conducted in accordance with PRISMA guidelines [67], with the completed PRISMA 2020 Checklist provided in S1 Checklist. The protocol has been registered with the international prospective register of systematic reviews (PROSPERO CRD42020183640 https://www.crd.york.ac.uk/prospero/display_record.php?ID= CRD42020183640).

### Search strategy and eligibility criteria

The search strategy aimed at identifying published studies that investigated the impact of workplace mental health interventions on aspects of mental health and wellbeing including stress, burnout and non-clinical symptoms of depression and anxiety in the construction industry. The search strategy can be found in S1 File. Unpublished 'grey literature' was not included. The search was performed using five databases on May 28, 2020, and updated on May 16, 2022: Academic Search Complete, PsycINFO, PubMed, Scopus and Web of Science and included studies published in English between January 2010 and May 2022 to reflect inter-vention efforts in the modern construction workplace. The search strategy was developed in an iterative process using the Population, Intervention, Comparison, and Outcome (PICO) framework and in consultation with the subject librarian within the lead author's (BAG) insti-tution. The search strategy was composed of free text and controlled vocabulary terms for workers in the construction industry (e.g., carpenters, labourers, etc.), intervention type (e.g., workplace mental health promotion), and outcomes (e.g., stress, burnout, mental wellbeing, etc.) grouped together using Boolean operators in consultation with the subject librarian. The search strategy underwent review following the PRESS guidelines [68] by a second and inde-pendent, subject librarian in the collaborating author's (BA) institution. Both backward and forward citation chaining of all articles included in the full-text review stage were conducted to identify additional studies that may have met the search criteria but were not previously found in the search results.

### Inclusion and exclusion criteria

Studies were included if they reported organisational-level mental health promotion interven-tions targeted at workers and/or managers within construction companies. In a staged approach only studies with a control group and with a before- and at least one after measure-ment post intervention were deemed eligible in the first round of selection as they provide the most robust evidence. These included randomised controlled trials (RCTs), a study design with participants randomly assigned to an intervention group and a control group; cluster ran-domised controlled trials (cRCT), with groups of participants (e.g., organisations) being ran-domised to an intervention and a control group, and non-randomised controlled trials. In a second stage, uncontrolled before- and after- designs and uncontrolled quasi-experimental designs were included in the reviews. Complete inclusion and exclusion criteria can be found in Table 1. Primary outcomes included quantitative measurements of aspects of mental health and wellbeing, secondary outcomes included organisational outcomes such as absenteeism, especially if linked to health issues, and changes to the psychosocial work environment. As organisational-level interventions target the psychosocial work environment, these changes were deemed important intermediary effects of mental health promotion. Absenteeism was deemed as a general organisational outcome measure, that is sensitive to short-term and long-

**Table 1. Inclusion and exclusion criteria.**

| Criteria | Description | Inclusion | Exclusion |
|---|---|---|---|
| Population | Construction industry | 1. Workers and managers in the construction industry according to the NACE classification: construction of buildings, civil engineering and specialised construction activities.<br>2. Fully employed or sub-contracted workers.<br>3. Full-time or part-time workers.<br>4. Workers in companies of all sizes | 1. Mainly non-working populations (unemployed, retired, long-term sick leave).<br>2. Populations not working in construction.<br>3. Apprentices or workers in training.<br>4. Clinical populations with mental health disorders. |
| Intervention | Organisational-level mental health promotion intervention | 5. Organisational-level intervention aimed at improving workers' mental health and/or wellbeing or protecting workers from mental health symptoms or disorders, at the level of the organisation by changing aspects of the psychosocial work environment (e.g., organisational policies, leadership style, workplace culture, working conditions) or through systematic training of work-related competencies.<br>6. Interventions designed to or that involve mental health knowledge and awareness building in the organisation or programs to train managers to initiate workplace changes.<br>7. Multi-level interventions targeting organisational and individual changes. | 5. Individual-level interventions solely aimed at changing employees' individual coping skills or behaviour and not embedded into the organisation.<br>6. Health promotion not primarily targeted at mental but at physical health.<br>7. Mental health interventions not formally implemented in the workplace.<br>8. Interventions that solely target individuals with a defined mental health disorder or disease for treatment and referral.<br>9. Interventions that solely target return-to-work after absenteeism due to mental health difficulties.<br>10. Evaluations focussing exclusively on the economic effects of mental health interventions. |
| Comparison | Control group<br>Pre- and post-comparison | 7. All experimental study designs with a comparison group, including RCTs, cRCTs, controlled before- and after- designs and controlled quasi-experimental studies.<br>8. Uncontrolled pre- and post-intervention comparison designs. | 11. Observational study designs and study designs with a single measurement.<br>12. Studies using solely qualitative research methods. |
| Outcomes | Primary: Mental health and wellbeing | 9. Stress, burnout, non-clinical depression and anxiety symptoms, mental wellbeing measured by validated scales or validated physiological indicators. | 13. Clinical mental health outcomes: severe depression and anxiety, diagnosed mental health disorders, suicide, suicidal ideation.<br>Substance abuse. |
|  | Secondary: Organisational | 10. Absenteeism.<br>11. Psychosocial work changes, specifically work demands, control/influence, social support by peers and by supervisors/ managers measured by validated scales. | 14. Presenteeism, turnover intention, productivity, job satisfaction, culture, stigma. |

term mental health effects of the promotion of wellbeing and the prevention of stress and burnout [69].

## Study identification

One researcher conducted each search in the respective databases (CL or CO'B). Results were exported into Rayyan QCRI, a software application to facilitate study selection in systematic reviews [70]. Duplicates were eliminated with the use of the Rayyan duplicate detection feature and verified by one reviewer (CL). To ensure adequate understanding and consistency in application of the inclusion and exclusion criteria, a sample of 20 records were selected at random and their titles and abstracts were reviewed and rated as 'eligible for inclusion' or 'not eligible for inclusion' independently by five authors (CL, CO'B, JCS, BG, BA) with the Rayyan blinding feature enabled. The five authors then met to discuss their inclusion decisions and any discrepancies were discussed until unanimous agreement was reached. Subsequently, two reviewers (CL, CO'B) completed a blinded title and abstract review of a random sample of 25% of records for inclusion. Agreement between the reviewers was 98.3% with the discrepancies resolved through discussion and did not require the input of a third reviewer. The remaining 75% of records were then screened at the title and abstract level by one reviewer (CL or

CO'B). Blind screening of full-text articles was completed by two authors (CL and BG), who agreed on final inclusion and exclusion decisions.

## Data extraction

Data extraction for the articles after full-text review included the following and was independently cross-checked by a third reviewer: 1, author and year; 2, type of study design; 3, number of participants and demographics, including employment type; 4, number of control participants and demographics (initial and analysed); 5, intervention details; 6, number of sessions and length; 7, type of control; 8, length of follow-up; 9, relevant outcomes; 10, instruments applied to measure outcomes; 11, country; 12, mean and standard deviation of all study groups in the relevant outcomes at all assessment times to be analysed; and 13, size of the organisation(s). Where data were missing, incomplete or unclear, requests for additional information were sent to the corresponding study authors by email. Additionally, information on the size of the organisations participating in the included studies was retrieved via company websites or LinkedIn profile pages.

## Quality appraisal

The quality of each included study was appraised using the Quality Assessment Tool for Quantitative Studies (QATQS) scale [71], which assesses 6 areas: 1, selection bias; 2, design; 3, confounders; 4, blinding; 5, data collection method; and 6, withdrawals and drop-outs. Results were scored on a scale from 1 to 3, where 1 is considered methodologically strong, 2 moderate and 3 weak and then globally ranked as methodologically 'strong' = no weak ranking, as 'moderate' = one weak ranking, and as 'weak' with two or more weak rankings. All studies were blindly appraised for quality using the QATQS by two independent reviewers (CL and CO'B) with independent checking of a third assessor (BAG) and any disagreements were discussed between the reviewers and resolved.

## Data presentation, analysis, and synthesis

We conducted a narrative synthesis of the findings from the included studies, structured around the effectiveness of the mental health intervention programmes. Mean differences (pre- and post- intervention) with p-values were reported and adjusted regression coefficients or adjusted Odds ratios with 95% confidence intervals, if available, were tabulated, brought together, and summarized using narrative synthesis. Reported outcomes, that were not specified as primary or secondary outcomes for the purpose of this review were not included in the results table nor the synthesis. It was not possible to conduct a meta-analysis for the included studies due to the diversity of interventions and outcomes.

The narrative synthesis was guided by the aim to identify which types of interventions were effective or not effective to positively promote mental health and wellbeing and to protect mental health in the construction sector. In the evidence synthesis, we considered the magnitude of effect found in each study and the methodological quality of the study as rated by the QATQS global score for each study. The classification developed in a systematic review on organisational-level wellbeing interventions by Fox et al [63] was applied to the narrative synthesis of studies using four categories: scheduling interventions, job and task modification, relational and team dynamics interventions and participatory process interventions. In addition, the Integrated Workplace Mental Health Intervention approach [57,58] served as a framework for synthesis and summary according to (a) interventions for the promotion of mental wellbeing and (b) interventions for the protection of mental health problems. Interventions for the promotion of mental wellbeing included interventions that were designed to improve the positive aspects of work and workers' strengths and capacities. Interventions for

the protection of mental health and wellbeing included interventions to reduce work risks in the psychosocial work environment. The third element of the Integrated Intervention Approach for Workplace Mental Health, i.e., 'manage illness' representing tertiary-level prevention, was not addressed in this review.

## Results

### Included studies and study characteristics

A total of 1326 records were identified from five databases. An additional 35 records were identified through the citation chaining of records included in the full-text review. Following removal of duplicates, the title and abstract of 1129 records were reviewed for eligibility. 1090 records were excluded, and the full text was reviewed for 39 articles. Primary reasons for exclusion at full-text review were ineligible outcomes (n = 14) and ineligible study design (n = 10). The complete results and decisions of inclusion or exclusion at the full-text review can be found in S2 File. After applying all exclusion criteria, five articles from four studies were identified as eligible for inclusion in the review (see Fig 1).

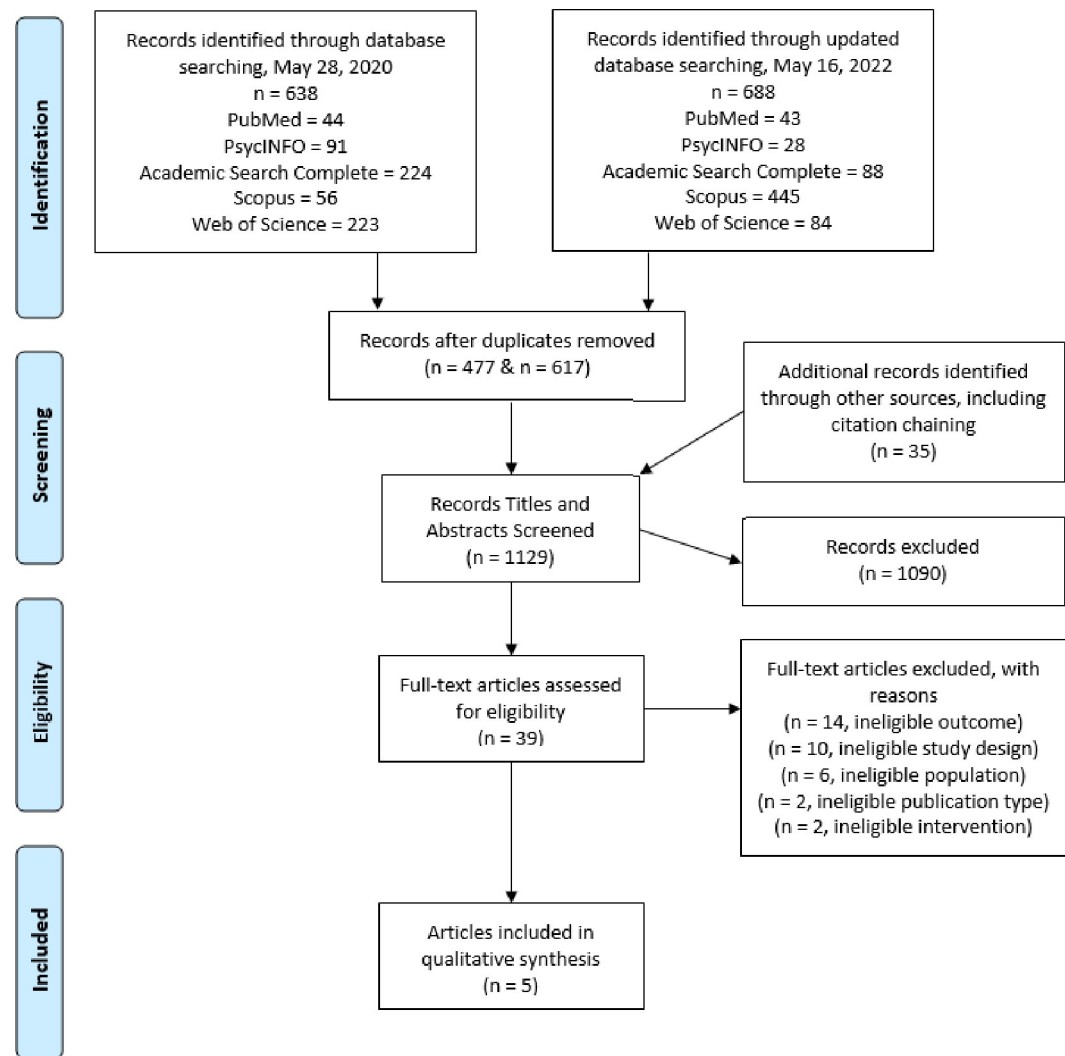

**Fig 1. PRISMA flow diagram.**

**Table 2. Study and sample characteristics.**

| First Author | Year | Country | Type of Study / Design | Intervention Group | | Control Group | | Employment type and occupations | Number and size of organisations |
|---|---|---|---|---|---|---|---|---|---|
| | | | | Sample Size (% Male) | Age (SD) | Sample Size (% Male) | Age (SD) | | |
| Anger [75] | 2018 | United States | Uncontrolled before and after study | Supervisors: 22 (90.9) Employees: 13 (69.2) | Supervisors: 39.2 (8.0) Employees: 37 (11.1) | N/A | N/A | Full-time employees and supervisors: carpenter, field, safety, and project engineers, site administrator, project coordinator or manager, health care market leader, and (senior) superintendent | 4 private sector organizations: 2 small to medium, 2 large |
| Elo [74] | 2013 | Finland | Non-randomised, controlled before and after study | Employees: 49 (67.3) | Employees: 44.7 (9.0) | Employees: 96 (84.3) | Employees: 43.9 (10.7) | Not reported; various trades | 1 large public sector organization |
| Guimaraes [76] | 2013 | Brazil | Uncontrolled before and after study | Employees: 5 (100) | Employees: 35.4 (9.91) | N/A | N/A | Full-time electricians | 1 large private sector organization |
| Oude Hengel [72,73] | 2012, 2013 | Netherlands | Cluster RCT | Employees: 171 (100) | Employees: 41.8 (12.7) | Employees: 122 (98.4) | Employees: 44.2 (12.7) | Bricklayers, carpenters, and others | 6 small to medium private sector construction companies |

We identified one cluster randomized controlled trial reported in two articles [72,73], one controlled before- and after study [74] and two uncontrolled before- and after studies [75,76] (Table 2). Anger at al. [75] presented a pilot study in preparation of a larger RCT. One study targeted the intervention (training to improve the psychosocial work environment) at supervisors only, with measurement of the changes in health outcomes in their subordinates [74], one study targeted the intervention at supervisors and shop-floor workers [75], and the remaining two studies targeted shop-floor workers only [72,73,76]. Total sample size across the four studies was 260 from a variety of occupations within construction. Sample sizes of the individual studies ranged between five [76], and 171 [72,73] individuals. Studies included predominantly male participants. SMEs were included in two studies [72,73,75].

Table 3 summarises the intervention characteristics of the four eligible studies.

Relational and team dynamics intervention was the focus of two studies [74,75], with an emphasis on supervisor personal growth and self-awareness development [74], and an emphasis on skills to establish sustainable and effective interactions with subordinates and to initiate organisational change supported by self-monitoring of skill application in practice [75].

The modification of the job and tasks featured in one study [76]. This study included the design and evaluation of a system of re-organising task allocation for live-line electricians to reduce physical and mental workload with consideration for circadian rhythm, the effect of heat and accumulation of fatigue over the work week.

Participatory process interventions were applied in two studies [73,75]. The intervention reported by Anger et al [75] was aimed at skill development for general healthy lifestyles with the use of scripted information, supervisory support and small group discussions. Oude-Hengel et al [72,73,77] applied an intervention to enable workers to modify their work techniques and methods and take adequate breaks to reduce physical and psychological work demands. This was supported by physiotherapists and based on work observations. In addition, empowerment training was delivered to workers to take a pro-active approach by taking responsibility for their own health and influence possible adverse working conditions by themselves.

**Table 3. Intervention characteristics of the included studies.**

| First Author | Year | Primary aim | Intervention details | *Duration of intervention* | Follow-up–measurement points |
|---|---|---|---|---|---|
| Anger | 2018 | To test the hypothesis that a Total Worker Health intervention could be implemented in the commercial construction industry and produce targeted positive impacts on Kirkpatrick's four levels of training evaluation. | Integrated approach to safety, health and wellbeing that includes computer-based training for supervisors, behavioural self-monitoring, and scripted training for supervisors and employees. | Varied (14 weeks in total). Supervisors: Two hours of training, five hours of tracking and increased interactions with employees that were expected to take about five minutes per employee per day. Employees: six hours weekly group meetings to discuss 'Get Healthier' topic using scripted cards and four hours for data collection. | Immediate |
| Elo | 2014 | To investigate the effect of a personal growth-oriented leadership intervention among line supervisor on subordinate wellbeing in a public sector construction organisation. | Leadership intervention which included creativity exercises, role playing, group discussions, short lectures on wellbeing and a participative stress management programme. | 7.5-day residential training | 2 years post intervention |
| Guimaraes | 2013 | With the assumption that circadian rhythms influence human performance, the work of live line electricians was reorganized and evaluated. | Optimized work system with consideration of (a) the circadian cycles and homeostatic processes; (b) the effect of heat on the biological clocks; and (c) the degree of physical and mental demands of the different performed tasks. | Two weeks: One-week traditional working schedule followed by one week of optimized working schedule (intervention) | Measurements during traditional work week versus optimized work week |
| Oude Hengel | 2012, 2013 | To evaluate the effectiveness of the worksite prevention program with an emphasis on preventing work-related musculoskeletal injury within a cluster randomized controlled trial. | A two-component prevention programme. Physical component: two individual training sessions by a physical therapist, and a rest-break tool. Mental health component: two interactive empowerment training sessions at the worksite aimed to help construction workers (i) take responsibility for their own health, (ii) discuss with colleagues about the responsibility for their own behaviour, and (iii) improve communication with the supervisor. | Over 6 months: 2 physical health and 2 mental health sessions; 30 minutes (physical health), 60 minutes (mental health). | 3-, 6-, and 12-months post intervention |

When classifying the studies according to the components of the Integrated Mental Health Promotion approach [58], we found that three studies targeted the aspect of 'Promoting mental health and wellbeing': Elo et al. [74] applied a growth-oriented approach for training supervisors to support their staff, Anger et al. [75] trained supervisors to engage in supportive interactions with employees, and Oude Hengel et al. [72,73] delivered 'empowerment training' to employees to gain influence in the organization. Two studies combined both approaches 'Promoting Mental Health and Wellbeing' and 'Protecting Mental Health' [72–74], however without explicit assessment of both, 'negative' mental health aspects and 'positive' wellbeing indicators. Two out of the four studies included mental health items within a broader quality of life measure (i.e., SF12), while one study specifically measured emotional exhaustion (Maslach Burnout Inventory). Apart from the study by Elo et al. [74], all studies had a strong focus on general health outcomes placing less focus on mental health outcomes. Although three of the four studies included in this review applied a growth-based approach for positive mental health promotion intending to strengthen or empower leaders and/or employees, none of the studies employed detailed outcome measures of positive mental wellbeing.

**Table 4. Summary of quality appraisal results.**

| First Author | Selection Bias | Design | Confounders | Blinding | Data Collection Method | Withdrawals and Drop-out | Global Rating |
|---|---|---|---|---|---|---|---|
| **Anger** | *Weak* | Moderate | *Weak* | Moderate | *Weak* | *Weak* | *Weak* |
| **Elo** | *Weak* | Moderate | Moderate | *Weak* | *Weak* | *Weak* | *Weak* |
| **Guimaraes** | Moderate | Moderate | *Weak* | *Weak* | *Weak* | Weak | *Weak* |
| **Oude Hengel** | Strong | Strong | Moderate | *Weak* | Moderate | Moderate | Moderate |

## Study quality

The overall methodological quality of one study [72,73] was rated as 'moderate', the remaining three studies as 'weak'. The cluster randomised control trial by Oude Hengel et al. was appraised as being strong on aspects of selection bias and study design and received a moderate rating for aspects of data collection methods, controlling for confounding and withdrawals/dropouts. Closer examination across the QATQS' six criteria revealed some differences between the studies. Table 4 provides a summary of the quality appraisal results. The remaining three included studies did not receive a strong rating across any of the six criteria, with only two of six criteria rated as 'moderate' for Guimaraes et al. [76] for Elo et al. [74], and for Anger et al. [75], respectively.

## Study outcomes

Tables 5 and 6 show the effectiveness for the four included studies for the primary and secondary outcomes. In relation to the primary outcomes (i.e., mental health and wellbeing), two studies used the mental health scale of the SF12 questionnaire, although neither showed a significant effect pre versus post intervention [73,75]. Furthermore, the Elo et al. study [74] used the emotional exhaustion scale from the Maslach Burnout Inventory to measure the effectiveness of supervisor training on mental health in subordinates with no significant intervention effects. Physiological indicators of stress were measured by Guimaraes [76] in a very small sample aiming to assess the effect of a task re-design intervention. Primarily, urinary adrenaline and nor-adrenaline levels significantly decreased in the intervention week with an optimized work system as compared to the baseline week with a traditional work system. While the ratio of catecholamines (nor-adrenaline/adrenaline) as an indicator for physical versus mental health load did not significantly differ pre- to post-intervention, the Q-Index (Catecholamine Ratio post- versus pre- intervention) significantly increased, suggesting a decline in mental health load in relation to physical load.

In relation to the secondary outcomes (psychosocial work environment, sickness absenteeism, and social support), Elo et al. [74] assessed workplace changes perceived by staff on a range of dimensions following a supervisor training programme. Subordinates of supervisors in the intervention group reported significant improvement in terms of flow of information, compared to subordinates in the control group. The results for work climate were inconclusive as work climate significantly decreased in the control group while it remained stable in the intervention group. None of the other psychosocial work characteristics significantly changed. Sickness absenteeism was measured in one study only [73], which showed a decline in long-term absences of 5 days, however, this result was not statistically significant. Results for social support at work, overall, between co-workers, and from supervisors were also not statistically significant across three follow-up measurement points [72].

## Discussion

### Summary of main results

This systematic review aimed to assess recent scientific evidence on the effectiveness of organisational-level mental health interventions on non-clinical mental health outcomes and mental

**Table 5. Results of the effectiveness of mental health promotion interventions: Primary outcomes.**

| First author | Outcome/Measurement | Scale range | Intervention group | | | Control group | | | Measure of effect (95% CI) |
|---|---|---|---|---|---|---|---|---|---|
| | | | N | Pre-intervention Mean (std) | Post-intervention Mean (std) | N | Pre-intervention Mean (std) | Post-intervention Mean (std) | |
| Anger | Mental Health/SF12 | 0–100 | 35 | 51.1 (6.6) | 51.1 (7.4) | - | n/a | n/a | Mean difference = 0, t = 0.11 (df = 34) Cohen d = 0 |
| Oude Hengel | Mental Health/SF12 | 0–100 | 155 | 55 (5.5) | 3 months: 54.6 (4.9) 6 months: 54.1 (7.2) 12 months: 54.5 (5.3) | 121 | 53.4 (7.7) | 3 months: 53.2 (7.0) 6 months: 53.5 (5.8) 12 months: 52.6 (7.5) | $\beta = 0.63^1$ (-1.07–2.33) $\beta = 0.12$ (-1.65–1.89) $\beta = 1.71$ (-0.08–3.49) Overall effect: $\beta = 0.80$ (-0.51–2.11) |
| Elo | Burnout/Emotional Exhaustion (Maslach Burnout Inventory) | 0–5 | 49 | 2.29 (1.49) | 2.40 (1.56) | 96 | 1.63 (1.16) | 1.56 (1.22) | $F = 0.80$, $p = 0.37^2$ |
| Guimaraes | Pulse during work | 17–39.5 | 5 | 21.4 | 24.6 | n/a | n/a | n/a | Pre-intervention versus intervention week p = 0.26³ |
| | Urinary nor-adrenaline | 22–54 | 2 | 41 | 27.25 | | | | Pre-intervention versus intervention week: p = 0.04³ |
| | Urinary adrenaline | 10–23 | 2 | 19.75 | 11.9 | | | | Pre-intervention versus intervention week: p = 0.02³ |
| | Catecholamine ratio (nor-adrenaline/adrenaline) | 1.85–3.3 | 2 | 2.3 | 2.61 | | | | P = 0.46³ |
| | Q-Index (Catecholamine ratio post intervention/catecholamine ratio pre intervention | 0.83–1.85 | 2 | 0.975 | 1.835 | | | | P = 0.01³ |

[1] Multilevel analysis in which the clusters time, worker and company were taken into account, adjusted model corrected for age and education. A positive $\beta$ means a higher mental health status of the intervention group compared to the control group.

[2] Analysis of variance with repeated measures: Group x time interaction.

[3] Analysis of variance with repeated measures: Worker, period, week. Only the effects for week are shown.

wellbeing in the construction industry in general and in SMEs in particular. This review identified a low number of studies on this topic with only one study of moderate methodological quality showing a substantial but non-significant decline in sick leave days [73] and no significant effect on general mental health. Results of the remaining 'weak-quality' studies included a significant effect of work redesign in short-term physiological stress parameters, however further research would be required to investigate the changes in physiological stress parameters over a longer period of time and their impact on long-term mental health [76]; a significant perceived improvement of flow of information after supervisor training [74]; no significant effect on general mental health [75] and no significant effect on a range of psychosocial work factors and on emotional exhaustion [74]. None of the four studies and interventions had an explicit focus on mental health, which may have contributed to the limited evidence for mental health and mental wellbeing in this review.

The evidence found in the four studies was not sufficient to develop specific recommendations for the design of organisational-level mental health promotion programmes in the construction industry and more specifically for SMEs. However, we identified two recently published protocol papers outlining the design of controlled studies to evaluate participatory

**Table 6. Results of the effectiveness of mental health promotion interventions: Secondary outcomes.**

| First author | Outcome/Measurement | Scale range | Intervention group | | | Control group | | | Measure of effect p-value |
|---|---|---|---|---|---|---|---|---|---|
| | | | N | Pre-intervention mean (std) | Post intervention means (std) | N | Pre-intervention mean (std) | Post- intervention mean (std) | |
| Elo | Psychosocial work characteristics (Healthy Organisation Questionnaire) | | | | | | | | |
| | Job demands | 1–5 | 49 | 3.49 (0.86) | 3.17 (0.93) | 96 | 3.12 (0.81) | 2.96 (0.86) | F = 0.98, p = 0.32[1] |
| | Job control | 1–5 | 49 | 2.99 (0.81) | 3.14 (0.80) | 96 | 3.18 (0.69) | 3.13 (0.69) | F = 2.61, p = 0.11[1] |
| | Information flow | 1–5 | 49 | 2.28 (0.81) | 2.55 (0.87) | 96 | 2.73 (0.85) | 2.69 (0.86) | F = 4.86, p = 0.03[1] F = 3.99, p = 0.05[2] |
| | Work climate | 1–5 | 49 | 2.87 (0.78) | 2.97 (0.94) | 96 | 3.45 (0.88) | 3.21 (0.75) | F = 7.61, p = 0.007[1] F = 5.61, p = 0.02[2] |
| | Support from supervisor | 1–5 | 49 | 3.36 (0.78) | 3.25 (0.94) | 96 | 3.65 (0.76) | 3.48 (0.85) | F = 0.10, p = 0.75[1] |
| | Feedback from supervisor | 1–5 | 49 | 2.84 (0.71) | 2.86 (0.91) | 96 | 3.17 (0.86) | 2.91 (0.85) | F = 3.03 p = 0.08[1] |
| | Justice of leadership | 1–5 | 49 | 2.96 (1.12) | 3.02 (1.09) | 96 | 3.71 (1.02) | 3.54 (1.01) | F = 2.05, p = 0.15[1] |
| Oude Hengel | Sick leave | | 171 | 6.8 (15.9) | N (%) | 122 | 6.4 (19.8) | N (%) | |
| | No or Short-term (<6 days) Long-term (> = 6 days) | | 170 | N (%) 128 (75) 42 (25) | 6 months: 139 (82) 30 (18) | 119 | N (%) 99 (83) 20 (17) | 6 months: 90 (76) 29 (24) | OR = 0.49 (0.17–1.20)[3] |
| | No or short-term (<6 days) Long-term (> = 6 days) | | 148 | | 12 months 113 (76) 35 (24) | 111 | | 12 months: 78 (70) 33 (30) | OR = 0.40 (0.15–1.57)[3] |
| | | | | | | | | | Overall effect: OR = 0.44 (0.13–1.26)[3] |
| Oude Hengel | Overall Social Support (Job Content Questionnaire) | 8–32 | 171 | 24.3 (2.5) | 3 months: 24.2 (2.5) 6 months: 25.5 (2.5) 12 months: 23.9 (2.5) | 122 | 24.0 (3.4) | 3 months: 24.2 (3.1) 6 months: 24.2 (3.2) 12 months: 24.0 (2.9) | β = 0.02 (-0.61–0.65) β = 0.25 (-0.40–0.90) β = -0.20 (-0.56–0.45) Overall effect: β = 0.03 (-0.39–0.46) |
| | Co-Worker Support | 4–16 | 171 | 12.4 (1.4) | 3 months: 12.3 (1.2) 6 months: 12.3 (1.4) 12 months: 12.2 (1.3) | 122 | 12.2 (1.7) | 3 months: 12.3 (1.5) 6 months: 12.3 (1.6) 12 months: 12.2 (1.4) | β = -0.02 (-0.33–0.30) β = 0.03 (-0.29–0.35) β = -0.02 (-0.35–0.30) Overall effect: β = 0.00 (-0.21–0.20) |
| | Supervisor Support | 4–16 | 171 | 12.0 (1.7) | 3 months: 11.9 (1.6) 6 months: 12.1 (1.7) 12 months: 11.7 (1.7) | 122 | 11.8 (2.0) | 3 months: 11.9 (1.9) 6 months: 11.8 (2.0) 12 months: 11.7 (1.8) | β = 0.07 (-0.34–0.48) β = 0.27 (-0.15–0.69) β = -0.09 (-0.51–0.33) Overall effect: β = 0.09 (-0.18–0.36) |

[1] Analysis of variance with repeated measures, group x time interaction–unadjusted models.

[2] Analysis of variance with repeated measures and adjustment for age, gender, education, type of work (white-collar/blue-collar) and subordinate participation in the organisation's stress management programme (days).

[3] Controlled for age and education.

mental health promotion programmes in construction workers [45,78]. The results of these planned trials may add to the evidence base in due course.

The interventions and the mental health outcomes in the four included studies were very heterogeneous, therefore the evidence did not allow for a meaningful summary by intervention type or outcome. Nevertheless, important lessons can be learned from the review considering the strengths and methodological shortcomings of the four studies to inform a future research agenda for designing and evaluating mental health interventions in the construction industry.

## Components of a future research agenda

**Develop and evaluate mental health interventions specific to the working situation in the construction sector.** Only one study included in this systematic review addressed the specific psychosocial working conditions of construction workers by using work task reallocation [74], while the other three studies were mostly generic in content, approach, and delivery. Tailoring the intervention more specifically to the workplace may be particularly relevant for the construction industry. As this industry exhibits a range of distinct workplace risks that can affect mental health and distinct implementation challenges due to the nature of its work. Previous research conducted in other sectors showed that a good contextual fit of the intervention activities and their implementation contributes to the success of the intervention implementation [79,80]. Management buy-in, participation of workers and integration of the intervention into the organisational context have been identified as key facilitators [81–83]. Co-creation of interventions has been encouraged to achieve context fit and buy-in of organisations [84,85] with a recent application to construction workers' mental health [82]. This strategy involved workers, managers and other stakeholders to define mutual goals, intervention activities and implementation approaches. Process evaluation showed promising results in relation to high involvement and ownership, development of trust in the intervention and a good contextual fit of the intervention [82]. Other participatory approaches specifically developed to target blue-collar populations [80] built on learning-by-doing principles with a focus on local practice, exchange of real-life experience and starting from real problems of the enterprise with recognition of achievements. While the principles of participatory design of interventions have been developed for general application, they still need to be tested for the application in the construction sector applying process and outcome evaluation.

**Develop and evaluate mental health interventions specific to the gender distribution and work culture of construction work.** Construction has been characterised as a male-dominated industry with a prevailing masculine culture associated with high mental health stigma, elevated levels of mental health shame and reduced help-seeking [35,50,86–88]. Research has consistently shown men to be less likely to participate in workplace health promotion programs in general [89], especially in terms of mental health interventions [90]. Reviews on the effectiveness of mental health interventions in male-dominated workplaces concluded that gender-sensitive health promotion approaches would be needed to address the unique issues in male-dominated workplaces, such as addressing gender-specific roles in disclosing mental health problems, activity-based programmes that were not seen as 'therapy' and peer-support oriented interventions including peer-to-peer outreach activities to tackle social isolation of apprentices in dispersed worksites [89,91,92]. One promising approach is the multi-component prevention and early intervention programme, MATES in Construction. Process evaluation has revealed the main perceived programme strengths. Construction workers positively noted that the programme was built into the culture of the construction industry so that they could relate to the mental health issues and identify with the addressed problems, they felt more confident in initially talking to their own trained work mates than medical

experts and experienced a built sense of camaraderie and high visibility of the trained peers on the dispersed work sites [36]. While the design principles of the MATES in Construction programme are highly relevant for research and practice in mental health promotion, the published evaluation studies of this approach focus on preventing suicide and do not include stress, burnout, and depressive and anxiety symptoms as outcomes, and therefore these studies were not eligible for inclusion in this review.

**Allow for complexity of intervention and multi-level design.** Organisational-level interventions tend to be complex with a range of intervention components and multiple levels targeted within organisations, including managers, supervisors, teams, and shop floor workers. They are usually difficult to standardise owing to local differences in company circumstances [79,93,94]. While clinical RCT designs have been considered the Gold Standard for study design, the updated Medical Research Council (MRC) Framework for the Development and Evaluation for Complex Interventions to Improve Health may be more suitable [95,96]. Guidance based on this framework promotes a more flexible intervention and evaluation approach tailored to local circumstances, designed with a range of short-term and long-term evaluation outcomes to adequately measure the complexity of the intervention. One of the major keys for designing complex interventions includes the formulation of theoretical models of how the different intervention components are expected to work and how their putative effects vary at different levels. The use of this framework helps to understand not only IF an intervention works but also HOW it works and may provide information on why it did not work.

**Develop a programme theoretical framework.** An explicit programme theory is necessary with specification of the expected short-term and long-term outcomes at different levels of the organisation, intermediary outcomes, and mechanisms of change to guide the intervention and evaluation and to avoid chance and error strategies. None of the studies included in our systematic review, spelled out such a model; only one study took multi-level variations into account by performing multi-level analyses [73]. In the context of workplace interventions, scholars have criticised that the effectiveness of organisational- or group-level interventions has predominantly been evaluated by investigating outcomes at the individual and not at the group level. However, organisational-level interventions usually target organisational units or groups of workers rather than individual workers. It has been suggested that this mismatch may also be responsible for the often weak and inconsistent results of the evaluation of organisational-level interventions [94] as also evident in our review. A forthcoming publication of a Theory of Change, developed by the MENTUPP Consortium will address this gap.

**Include a range of mental health and wellbeing outcomes.** Intervention research should measure a range of mental health and wellbeing outcomes. Our review showed that studies applied limited measurement of mental health and mainly focussed on pathology-oriented concepts of mental health problems, such as burnout and distress, without detailing positive aspects of mental health and wellbeing and recognising the multi-dimensional nature of wellbeing recommended by a newly developed conceptual framework for construction interventions [55].Interventions focussed on strengthening the positive aspects of work (e.g. improving social support, enhancing influence at work) to pursue positive outcomes (e.g. wellbeing, resilience) are very relevant in workplace mental health promotion [57,58], as positive aspects of work can support employees in dealing with stressful situations and can themselves contribute to positive wellbeing outcomes. By not measuring positive mental health outcomes, some improvements due to the intervention may remain unnoticed.

**Measure workplace changes as intermediary outcome.** Workplace changes post intervention should be explicitly measured as they are an important intermediary outcome in the causal chain. Organisational-level interventions typically target the improvement of the psychosocial work environment by job or task redesign, cultural change, and policy with the aim

to achieve long-term changes in mental health and wellbeing. Measurement of work environment variables would help to understand how an intervention works and which particular work characteristics should be targeted by the intervention. Two studies included in our systematic review assessed the changes in psychosocial work characteristics before and after the intervention, however without linking these characteristics to mental health outcomes [72,74].

**Develop complex strategies to evaluate the impact of leader training on workplace changes and staff mental health.** The complexity and methodological challenges of organisational-level mental health interventions are particularly evident in studies applying supervisor training and measuring mental health outcomes of this training on subordinates. Supervisor training is not only aimed at changing outcomes in individual supervisors, but it is also assumed that trained supervisors initiate workplace changes and change their behaviour towards staff hereby positively influencing the mental health and wellbeing of groups of staff. Two of the included studies [74,75] applied leadership training, an intervention component that is commonly considered relevant in the context of organisational-level interventions with supervisors and managers commonly being in the position to influence the design of psychosocial working conditions. While a meta-analysis of 10 controlled studies found indications of positive training effects on self-reported managers' knowledge, attitudes and giving support to employees, with mental health problems the on employees' mental health remains preliminary due to very few studies [97]. Evaluating the effectiveness of supervisor training on a distant mental health outcome in employees is complex and requires a multi-level and complex strategy for analysis. While the included study by Elo et al. [74] showed that employees reported positive significant changes in information flow pre- to post-intervention, multi-level analyses were not applied and the association of these changes with mental health outcomes remains to be demonstrated.

## Strengths and limitations of this review

To our knowledge this is the first systematic review appraising the effectiveness of organisational-level workplace mental health interventions in the construction industry. The strengths of this review are the rigour applied to the literature searches (five major data bases, extensive and detailed search terms, peer review of search strategy by experts, hand searching within the retrieved full text reference lists), the internal quality assessment of the interrater reliability of the reviewers, the rigorous quality assessment of the studies and the detailed synthesis of the outcomes. However, there were also certain limitations of this effectiveness review. Although our search strategy was comprehensive and included five major scientific databases, we may have missed important publications. This was somewhat mitigated by hand-searching the reference lists of all included studies for additional eligible publications. The review was limited to studies published in English and did not include the 'grey literature' to allow for the scientifically robust and peer-reviewed evidence. We may have missed relevant evidence from non-published studies and cannot exclude that our findings were influenced by publication bias. Finally, as the search was limited to studies published since 2010, the review may have missed relevant evidence that preceded this period. However, because of rapidly changing working conditions also in the construction industry, we decided to focus on recent studies with the meaningful cut-off point of 2010 to include studies potentially conducted in the height of the worldwide economic depression and published after 2009 and also encompasses the post-depression period.

## Conclusions

Although based on a low number of studies with scarce evidence, this systematic review exemplified a range of organisational-level mental health intervention approaches including

relational and team dynamics interventions with supervisor and shop floor workers, modification of job tasks with changed task allocation procedures, and participatory process interventions to empower and enable workers and supervisors to make a change in the organisation. These examples could be taken as stepping stones to develop, refine and scientifically evaluate organisational-level mental health interventions combined with implementation guidance specific to the challenges of the construction work environment.

In keeping with the general mental health perspective, this review focussed on a selected range of mental health and mental wellbeing outcomes. Further reviews may add synthesised evidence for the wider dimensions of wellbeing, such as job satisfaction, life satisfaction, work engagement, and work-life balance. With strict inclusion criteria for the methodological quality of the individual studies, the low number of papers included in this review does not reflect the multitude of initiatives and programmes established in practice. However, some programmes appear to lack robust scientific evaluation. There is a rich opportunity for scientific effectiveness and process evaluation of existing and future workplace mental health programmes to determine whether they actually result in mental health improvements, which programme elements are the most effective and which approaches can be best implemented into the construction work environment. Multi-level approaches for the design of future studies are desirable to overcome limitations of previous studies and would also greatly inform organisational-level mental health interventions in other sectors.

## Supporting information

**S1 Checklist. Completed PRISMA 2020 checklist.**
(DOCX)

**S1 File. Sample search strategy.**
(PDF)

**S2 File. Outcomes of full-text review.**
(DOCX)

## Acknowledgments

We acknowledge the contributions of the following MENTUPP Consortium members in the writing of this manuscript: Ainslie O'Connor, Ana Moreno-Alcázar, Andia Meksi, Andras Szekely, Anthony LaMontagne, Ariel Como, Arilda Dushaj, Asmae Doukani, Azucena Justicia, Bridget Hogg, Chantal Van Audenhove, Charlotte Paterson, Chris Lockwood, David McDaid, Doireann Ni Dhalaigh, Dooyoung Kim, Eileen Williamson, Eva Zsak, Eve Griffin, Fotini Tsantilla, Genc Burazeri, Gentiana Qirjako, Grace Davey, Gyorgy Purebl, Hanna Reich de Paredes, Jaap van Weeghel, Joe Eustace, Joseph Kilroy, Juliane Hug, Kairi Kolves, Karen Mulcahy, Karen Michell, Katherine Thomson, Laura Cox, Luigia D'Alessandro, Mallorie Leduc, Mónika Ditta Tóth, Naim Fanaj, Nicola Reavley, Peter Trembeczky, Reiner Rugulies, Ruth Benson, Saara Rapeli, Sarah Ihinonvien, Sarita Sanches, Sevim Mustafa, Sharna Mathieu, Stefan Hackel, Tanya King, Ulrich Hegerl, Vanda Scott, Victor Pérez, and Wendy Orchard.

## Author Contributions

**Conceptualization:** Birgit A. Greiner, Caleb Leduc, Johanna Cresswell-Smith, Reiner Rugulies, Birgit Aust.

**Data curation:** Caleb Leduc.

**Formal analysis:** Birgit A. Greiner, Caleb Leduc.

**Funding acquisition:** Birgit A. Greiner, Kristian Wahlbeck, Benedikt L. Amann, Paul Corcoran, Margaret Maxwell, Ella Arensman, Birgit Aust.

**Investigation:** Caleb Leduc, Cliodhna O'Brien, Birgit Aust.

**Methodology:** Birgit A. Greiner, Caleb Leduc, Birgit Aust.

**Supervision:** Birgit A. Greiner, Birgit Aust.

**Writing – original draft:** Birgit A. Greiner, Caleb Leduc, Cliodhna O'Brien, Birgit Aust.

**Writing – review & editing:** Birgit A. Greiner, Caleb Leduc, Cliodhna O'Brien, Johanna Cresswell-Smith, Reiner Rugulies, Kristian Wahlbeck, Kahar Abdulla, Benedikt L. Amann, Arlinda Cerga Pashoja, Evelien Coppens, Paul Corcoran, Margaret Maxwell, Victoria Ross, Lars de Winter, Ella Arensman, Birgit Aust.

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
