## [Decision Letter · Decision Letter 0]

22 Aug 2022

PONE-D-22-20510The effectiveness of organisational-level workplace mental health interventions on mental health and wellbeing in the construction industry: a systematic review and recommended research agendaPLOS ONE

Dear Dr. Greiner,

Thank you for submitting your manuscript to PLOS ONE. After careful consideration, we feel that it has merit but does not fully meet PLOS ONE’s publication criteria as it currently stands. Therefore, we invite you to submit a revised version of the manuscript that addresses the points raised during the review process.

Refer to comments raised by both reviewers to help improve the quality of the manuscript.

We look forward to receiving your revised manuscript.

Kind regards,

Haruna Musa Moda

Academic Editor

PLOS ONE

Journal Requirements:

Additional Editor Comments:

Dear Authors,

Your manuscripts review has now been reviewed and will recommend further review based on both reviewers comments.

Best regards

Reviewers' comments:

Reviewer's Responses to Questions

**Comments to the Author**

1. Is the manuscript technically sound, and do the data support the conclusions?

Reviewer #1: Yes

Reviewer #2: Partly

2. Has the statistical analysis been performed appropriately and rigorously? 

Reviewer #1: Yes

Reviewer #2: Yes

3. Have the authors made all data underlying the findings in their manuscript fully available?

Reviewer #1: Yes

Reviewer #2: Yes

4. Is the manuscript presented in an intelligible fashion and written in standard English?

Reviewer #1: Yes

Reviewer #2: Yes

5. Review Comments to the Author

Reviewer #1: This is generally a good work and merits publication in the journal for international readership. The author(s) have demonstrated adequate understanding of concepts and theories around the subject matter and the Review Methodology was clearly and logically articulated.

The author(s) should consider Lines 329 – 332 on page 17. The message is not clear and needs to be reviewed for readability.

On pages 20 and 21, the reporting of the reviewed studies is too detailed. Author(s) should reduce the details as interested readers can refer to the paper. Author(s) should focus on the effectiveness of the interventions being reviewed in this work and make more synthesis to bring out important lessons to the readers

The paper needs to be proofread for a few grammatical and typographical issues

Reviewer #2: 1. I believe the tittle is not complete. There is need to identify the primary subject of the research. Either workers, employees, or professionals in the construction industry.

2. What does RCTs, cRCTs, mean. There is not clear description of the terms in the text.

3. The second paragraph of the introduction section (line 95-106) discussed sucide as a consequence of mental; health challenge. I believe discussing general consequences of mental health challenges would provide more insight as opposed to narrowing it down to just suicide.

4. The section "Workplace mental health promotion in the construction sector" has only one paragraph which lacks structure. it contains 3 distinct topics which could constitute 3 different paragraphs. it is recommended to discuss each topic in detail.

a. limited number of studies focus on mental health in the sector

b. barriers to successful mental health intervention programs in the sector

c. various intervention programs often used in the sector.

5. is there any record that the interventions have been applied to some extent in the construction industry from previous studies or observations? if Yes, such accounts should be duly reported to provide premise to the arguments in this study.

6. Line 195-199; statement is not clear. As it is, it posits that the non-clinical challenges are outcomes of the interventions. i.e the interventions result in the non-clinical mental health challenges against the contrary. is this correct?

7. Line 200-203; is the focus only on positive impact or does it include negative impact as well?

8. There seems to be contradictions in the scoping of the study. was the focus mainly on interventions in SME organisations or generic organisations in the sector regardless of size? There might be evidence of such interventions in larger construction environments than in SMEs. The scope of the study needs to be clearly defined.

9. What is the objective rationale behind the time frame for the search (2010-2022). is there evidence that supports change in workplace dynamics every decade? why not 2 decades? or half a decade?

10. in Table 1, number the inclusion and exclusion criteria. i.e provide SN for each item under the inclusion and exclusion columns.

11. What were the search keywords used ? clearly state in the study identification section.

12. What is Rayyan QCRI?

13. I can understand the very low results (only 5 articles out of 1129) considering the strict exclusion criteria. but what were the search keywords used? justify the low outcome.

14. The conclusion is scanty and misleading. Needs to be improved to align with the study findings and provide objective inferences. E.g., there maybe scarcity of studies on the application and effectiveness of mental health interventions in the construction sector but it is not conclusive of the lack of actual application of interventions in the industry.

6. PLOS authors have the option to publish the peer review history of their article (what does this mean?). If published, this will include your full peer review and any attached files.

Reviewer #1: **Yes: **

Reviewer #2: No

---

## [Author Response · Author response to Decision Letter 0]

6 Oct 2022

Dear Chief Editor,

Thank you for inviting us to resubmit our revised manuscript. We would also like to thank both reviewers for their careful reading and their helpful comments, which helped us to improve the quality of the manuscript. 

Please find our detailed commentaries in response to the respective reviewer comments attached.

Please let me know whether you need more information.

Kind regards,

Dr Birgit Greiner

Reviewer #1: 

1. This is generally a good work and merits publication in the journal for international readership. The author(s) have demonstrated adequate understanding of concepts and theories around the subject matter and the Review Methodology was clearly and logically articulated.

We thank the reviewer for the complimentary words.

2. The author(s) should consider Lines 329 – 332 on page 17. The message is not clear and needs to be reviewed for readability.

We cleaned this sentence and cut it into several sentences and hope that the information is clearer now. It now reads as following: 

“Total sample size across the four studies was 260 from a variety of occupations within construction. Sample sizes of the individual studies ranged between five (1), and 171 (2,3) individuals. Studies included predominantly male participants. SMEs were included in two studies (2–4).”

3. On pages 20 and 21, the reporting of the reviewed studies is too detailed. Author(s) should reduce the details as interested readers can refer to the paper. Author(s) should focus on the effectiveness of the interventions being reviewed in this work and make more synthesis to bring out important lessons to the readers.

We thank the reviewer for this suggestion. We used the categorisation developed in a systematic review of organisational-level wellbeing intervention types by Fox et al to synthesise the interventions, shortened the presentation of the characteristics of the individual studies in the main text and also synchronised the information with the details provided in Tables 2 and 3. 

4. The paper needs to be proofread for a few grammatical and typographical issues

The manuscript was proof-red. We made a few grammatical edits and corrected the typos throughout the article. 

Reviewer #2: 1. 

I believe the title is not complete. There is need to identify the primary subject of the research. Either workers, employees, or professionals in the construction industry.

We now changed the title to: “The effectiveness of organisational-level workplace mental health interventions on mental health and wellbeing in construction workers: a systematic review and recommended research agenda”

2. What does RCTs, cRCTs, mean. There is not clear description of the terms in the text.

We introduced both acronyms, which are widely used in the scientific literature, with their first mentioning in the abstract and, again, with their first mentioning in the main text. 

We also added a short definition of both study design types to the inclusion and exclusion criteria presented in the methods part. This now reads as:

“In a staged approach only studies with a control group and with a before- and at least one after measurement post intervention were deemed eligible in the first round of selection as they provide the most robust evidence. These included randomised controlled trials (RCTs), a study design with participants randomly assigned to an intervention group and a control group; cluster randomised controlled trials (cRCT), with groups of participants (e.g., organisations) being randomised to an intervention and a control group, and non-randomised controlled trials. In a second stage, uncontrolled before- and after- designs and uncontrolled quasi-experimental designs were included in the reviews. Complete inclusion and exclusion criteria can be found in Table 1.”

3. The second paragraph of the introduction section (line 95-106) discussed suicide as a consequence of mental; health challenge. I believe discussing general consequences of mental health challenges would provide more insight as opposed to narrowing it down to just suicide.

We agree and discussed mental health in the wider context of common mental health disorders, work ability and safety. Please refer to the revised manuscript.

4. The section "Workplace mental health promotion in the construction sector" has only one paragraph which lacks structure. It contains 3 distinct topics which could constitute 3 different paragraphs. it is recommended to discuss each topic in detail.

a. limited number of studies focus on mental health in the sector

b. barriers to successful mental health intervention programs in the sector

c. various intervention programs often used in the sector.

5. Is there any record that the interventions have been applied to some extent in the construction industry from previous studies or observations? if Yes, such accounts should be duly reported to provide premise to the arguments in this study.

We thank the reviewer for these helpful suggestions and separated the text into 3 paragraphs with the following structure:

a. Various intervention programmes in this sector

b. Barriers to successful implementation of mental health promotion programmes in this sector

c. Limitations of existing approaches and new theoretical developments

Please refer to the changes in the revised manuscript. We added detail to the presentation of common programmes in this sector to give a more comprehensive account of the various initiatives, studies and observations (scientifically published and unpublished) in construction. We also moved the presentation of the MATES in Construction programme from the discussion section to this section. We added a section on limitations of current programmes and briefly highlighted new theoretical development. This last part is now more closely linked to the following section of the manuscript and builds a stronger argument for focussing on organisational-level interventions.

6. Line 195-199; statement is not clear. As it is, it posits that the non-clinical challenges are outcomes of the interventions. i.e the interventions result in the non-clinical mental health challenges against the contrary. is this correct?

Thank you for pointing out our misleading use of language. We clarified the issue. This sentence now reads: 

“While previous reviews conducted by the MENTUPP Consortium have reviewed the literature for effects of interventions addressing clinical mental health disorders across all industries [45], this review focuses on the effectiveness of organisational-level interventions specific to the construction industry to reduce non-clinical mental health problems (i.e., stress, burnout, and moderately elevated depressive and anxiety symptoms), and enhance wellbeing.”

7. Line 200-203; is the focus only on positive impact or does it include negative impact as well?

We were specifically looking for the effectiveness of mental health interventions in reducing non-clinical mental health outcomes and in enhancing wellbeing. This does not exclude the discussion of studies that do not show an effect or even show an effect in the opposite direction. We rephrased our review questions to reflect this better and split the review question into 2 questions.

“The specific review questions are: (1) Are organisational-level mental health programmes effective in reducing stress, burnout, non-clinical depressive and anxiety symptoms, and in enhancing mental wellbeing in construction workers? (2) Are organisational-level mental health programmes effective in reducing stress, burnout, non-clinical depressive and anxiety symptoms and in enhancing mental wellbeing in SME construction workers?”

8. There seems to be contradictions in the scoping of the study. was the focus mainly on interventions in SME organisations or generic organisations in the sector regardless of size? There might be evidence of such interventions in larger construction environments than in SMEs. The scope of the study needs to be clearly defined.

The scope of the review was not limited by company size and literature searches did not exclude any studies by company size. The size of the investigated companies was documented at data extraction phase to allow for sub-analyses specifically for SMEs. We now added an explicit inclusion criterion (‘no limitation by company size’) to the list in Table 1. We also clarified this issue by splitting the review question into 2 questions (see response to comment 7). 

9. What is the objective rationale behind the time frame for the search (2010-2022). is there evidence that supports change in workplace dynamics every decade? why not 2 decades? or half a decade?

When defining our search period from 2010-2022, we carefully balanced several considerations. We were aiming at including studies conducted in the modern construction workplace but sufficiently going back in time to capture a relevant timeframe. The 2010 cut-off point is meaningful in the context of the worldwide economic recession 2008/09, which affected the construction industry very heavily. https://www.ilo.org/global/publications/world-of-workmagazine/articles/WCMS_115508/lang--en/index.htm

On a population level, workplace mental health can be negatively affected by economic recession. The selection of our search period reflects the period of studies potentially conducted in the height of depression and published after 2009 and also encompasses the post-depression period. We added a brief justification of the limited time period to the methods section.

10. in Table 1, number the inclusion and exclusion criteria. i.e., provide SN for each item under the inclusion and exclusion columns.

We numbered the inclusion and exclusion criteria.

11. What were the search keywords used? Clearly state in the study identification section.

We used a comprehensive set of search keywords and wildcards linked with Boolean operators. The search strings together with an example search are provided in the additional online material with a cross-reference in the main text. We feel that the search strings are too complex to be presented in the main text. In order to enhance the visibility of the cross-reference to the additional material, we moved the reference to the beginning of the section entitled ‘Search strategy and eligibility criteria’.

12. What is Rayyan QCRI?

Rayyan QCRI is a widely used tested software application to facilitate the screening of articles and management of references for systematic reviews. It features easy recognition and elimination of duplicate references and online collaboration of researchers working on a review, e.g., a blinding feature that allows two researchers to conduct eligibility screening independently from each other.

We added a description, a reference and clearer rationale for the use of Rayyan QCRI and a reference. This section now reads as follows:

“Results were exported into Rayyan QCRI, a software application to facilitate study selection in systematic reviews [70] . Duplicates were eliminated with the use of the Rayyan duplicate detection feature and verified by one reviewer (CL). To ensure adequate understanding and consistency in application of the inclusion and exclusion criteria, a sample of 20 records were selected at random and their titles and abstracts were reviewed and rated as ‘eligible for inclusion’ or ‘not eligible for inclusion’ independently by five authors (CL, CO’B, JCS, BG, BA) with the Rayyan blinding feature enabled.”

13. I can understand the very low results (only 5 articles out of 1129) considering the strict exclusion criteria. but what were the search keywords used? justify the low outcome.

This is an important aspect. Results of systematic reviews need to be judged by the scope of the search terms. We agree with the reviewer, that our inclusion criteria were strict and the scarcity of identified studies does not reflect the lack of actual application of interventions in the industry. We expanded the conclusion section to take account of this (Please also see our response to comment 14). 

Within our inclusion and exclusion criteria we feel that we applied a very thorough search with application of wide search terms, also including wildcards etc and captured a wide range of aspects. Please refer to the attached documentation for the search strategies. In addition, we also hand-searched the reference list of retrieved full-text articles. This makes us confident that we found the vast majority of published articles that meet our inclusion criteria. We expanded the strengths and limitations part in the discussion section to that regard. 

“The strengths of this review are the rigour applied to the literature searches (five major data bases, extensive and detailed search terms, peer review of search strategy by experts, hand searching within the retrieved full text reference lists), the internal quality assessment of the interrater reliability of the reviewers, the rigorous quality assessment of the studies and the detailed synthesis of the outcomes.”

14. The conclusion is scanty and misleading. Needs to be improved to align with the study findings and provide objective inferences. E.g., there may be scarcity of studies on the application and effectiveness of mental health interventions in the construction sector but it is not conclusive of the lack of actual application of interventions in the industry.

We thank the reviewer for this suggestion and expanded the conclusions to point out the mismatch between interventions in practice and published evaluation studies in the construction sector. This section now reads as follows:

“Although based on a low number of studies with scarce evidence, this systematic review exemplified a range of organisational-level mental health intervention approaches including relational and team dynamics interventions with supervisor and shop floor workers, modification of job tasks with changed task allocation procedures, and participatory process interventions to empower and enable workers and supervisors to make a change in the organisation. These examples could be taken as stepping stones to develop, refine and scientifically evaluate organisational-level mental health interventions combined with implementation guidance specific to the challenges of the construction work environment. 

In keeping with the general mental health perspective, this review focussed on a selected range of mental health and mental wellbeing outcomes. Further reviews may add synthesised evidence for the wider dimensions of wellbeing, such as job satisfaction, life satisfaction, work engagement, and work-life balance. With strict inclusion criteria for the methodological quality of the individual studies, the low number of papers included in this review does not reflect the multitude of initiatives and programmes established in practice. However, some programmes appear to lack robust scientific evaluation. There is a rich opportunity for scientific effectiveness and process evaluation of existing and future workplace mental health programmes to determine whether they actually result in mental health improvements, which programme elements are the most effective and which approaches can be best implemented into the construction work environment. Multi-level approaches for the design of future studies are desirable to overcome limitations of previous studies and would also greatly inform organisational-level mental health interventions in other sectors.”

---

## [Decision Letter · Decision Letter 1]

20 Oct 2022

The effectiveness of organisational-level workplace mental health interventions on mental health and wellbeing in construction workers: a systematic review and recommended research agenda

PONE-D-22-20510R1

Dear Dr. Greiner,

We’re pleased to inform you that your manuscript has been judged scientifically suitable for publication and will be formally accepted for publication once it meets all outstanding technical requirements.

Kind regards,

Haruna Musa Moda

Academic Editor

PLOS ONE

Additional Editor Comments (optional):

Dear Dr Greiner,

I am happy to recommend your manuscripts for acceptance based on the positive review received from both reviewers.

Congratulations.

Reviewers' comments:

Reviewer's Responses to Questions

**Comments to the Author**

1. If the authors have adequately addressed your comments raised in a previous round of review and you feel that this manuscript is now acceptable for publication, you may indicate that here to bypass the “Comments to the Author” section, enter your conflict of interest statement in the “Confidential to Editor” section, and submit your "Accept" recommendation.

Reviewer #1: All comments have been addressed

Reviewer #2: All comments have been addressed

2. Is the manuscript technically sound, and do the data support the conclusions?

Reviewer #1: Yes

Reviewer #2: Yes

3. Has the statistical analysis been performed appropriately and rigorously? 

Reviewer #1: Yes

Reviewer #2: Yes

4. Have the authors made all data underlying the findings in their manuscript fully available?

Reviewer #1: Yes

Reviewer #2: Yes

5. Is the manuscript presented in an intelligible fashion and written in standard English?

Reviewer #1: Yes

Reviewer #2: Yes

6. Review Comments to the Author

Reviewer #1: All issues raised in the previous review of the paper have been fully and satisfactorily addressed, and the paper merits immediate publication.

Reviewer #2: The authors have done a very good job in addressing all of the reviewer's comments. The article is acceptable in its current state, subject to editorial edits.

7. PLOS authors have the option to publish the peer review history of their article (what does this mean?). If published, this will include your full peer review and any attached files.

Reviewer #1: **Yes: **Dr. Mu'awiya Abubakar

Reviewer #2: **Yes: **Bello Mahmud Zailani

---

## [Editor Report · Acceptance letter]

8 Nov 2022

PONE-D-22-20510R1 

The effectiveness of organisational-level workplace mental health interventions on mental health and wellbeing in construction workers: a systematic review and recommended research agenda 

Dear Dr. Greiner:

I'm pleased to inform you that your manuscript has been deemed suitable for publication in PLOS ONE. Congratulations! Your manuscript is now with our production department. 

Kind regards, 

on behalf of

Dr. Haruna Musa Moda 

Academic Editor

PLOS ONE